# Competing multiple oxidation pathways shape atmospheric limonene-derived organonitrates simulated with updated explicit chemical mechanisms

Qinghao Guo[1], Haofei Zhang[2], Bo Long[3], Lehui Cui[1], Yiyang Sun[1], Hao Liu[1], Yaxin Liu[1], Yunting Xiao[1], Pingqing Fu[1] and Jialei Zhu[1,*]

1 Institute of Surface-Earth System Science, School of Earth System Science, Tianjin University, Tianjin, 300072, China;

2 Department of Chemistry, University of California, Riverside, California 92521, USA;

3 College of Materials Science and Engineering, Guizhou Minzu University, Guiyang 550025, China.

*Correspondence to: Jialei Zhu, Email: zhujialei@tju.edu.cn*

**Abstract.** Organonitrates (ON) are key components of secondary organic aerosols (SOA) with potential environmental and climate effects. However, ON formation from limonene, a major monoterpene with unique structure, and its sensitivity to oxidation pathways remain insufficiently explored due to the absence of models with explicit chemical mechanisms. This study advances the representation of limonene-derived ON formation by incorporating 90 gas-phase reactions and 39 intermediates across three oxidation pathways ($O_3$, OH, $NO_3$) into both a chemical box model and a global model. Box model sensitivity experiments revealed that competition among major oxidation pathways, coupled with the high yield of limonene-derived ON from $O_3$-initiated oxidation, leads to increased limonene-derived ON production when the $O_3$-initiated pathway is enhanced, whereas strengthening the OH- or $NO_3$-initiated pathways reduces ON formation. Compared to the box model, the global simulation exhibits stronger nonlinear responses and great spatiotemporal variability in limonene-derived ON formation across different oxidation pathways. This is primarily driven by the complex distribution of precursors and oxidants, as well as changing in dominate chemical pathways under various meteorological conditions. In the presence of the other two pathways, increasing the $O_3$- or $NO_3$-initiated oxidation pathway reduces the global limonene-derived ON burden by 19.9% and 17.3%, respectively, whereas enhancing the OH-initiated pathway increases it by 44.7%. limonene-derived ON chemistry developed in this study not only enhances the global model's ability to simulate ON formation evaluated through comparison with observations but also demonstrates an approach based on explicit chemical mechanisms that establishes a methodological framework for simulating the chemical formation processes of SOA.

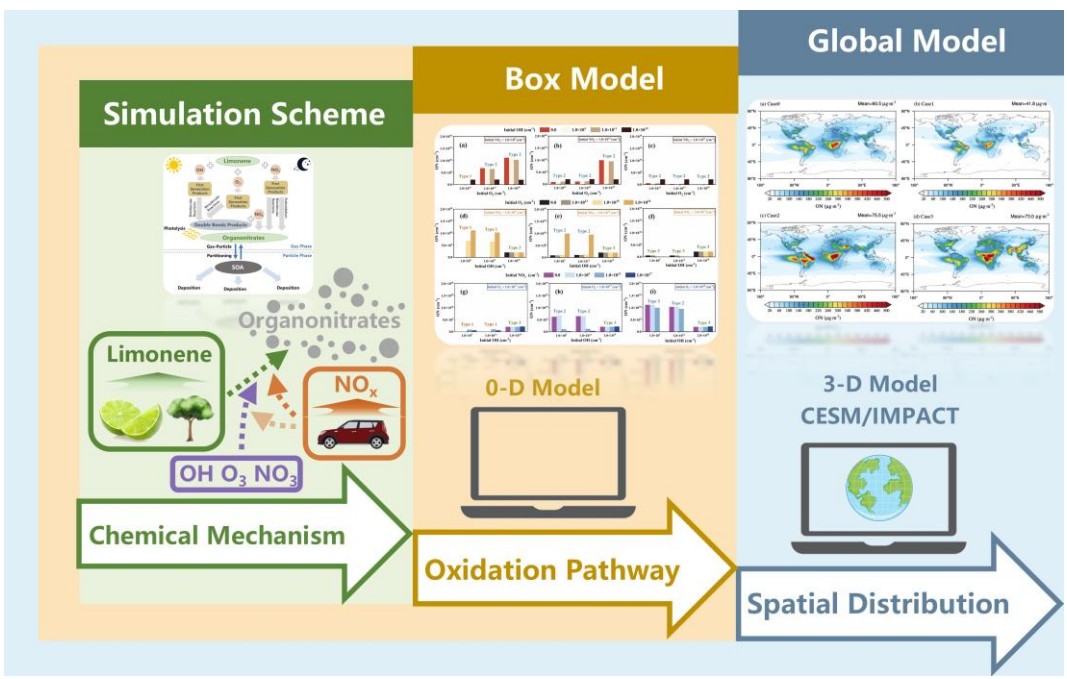

29

**1 Introduction**

Secondary organic aerosols (SOA) represent a substantial fraction of fine particulate matter and contribute to global public health risk, deterioration of air quality and climate change (Collaborators, 2024; Lelieveld et al., 2015; Tao et al., 2017). Among chemical constituents, organonitrates (ON) are of particular interest owing to their large fraction in SOA (5%-77%) (Farmer et al., 2010; Kiendler-Scharr et al., 2016). The rate of particulate ON formation contributes strongly to the rate of SOA formation at night, which emphasizes the important roles of particulate ON in ambient SOA (Guo et al., 2024). The nitrate group in ON would influence the physical and chemical properties of SOA, such as decreasing saturated vapor pressure of the product molecule (Capouet and Müller, 2006). ON are secondary compounds formed via the oxidation of volatile organic compounds (VOCs) in the presence of nitrogen oxides ($NO_x = NO + NO_2$), substantially influencing $NO_x$ cycling and formation of ozone ($O_3$) and HONO (Perring et al., 2013). In global scale, VOCs are mainly emitted from biogenic sources, while $NO_x$ are emitted from a wide variety of anthropogenic sources (Ng et al., 2017; Glasius and Goldstein, 2016). Therefore, a thorough investigation of ON is warranted to advance our understanding of interaction between biogenic and anthropogenic emissions.

The chemical formation mechanisms of ON are complex, hampering efforts to simulate and control SOA. In the daytime, hydroxyl radicals (OH) and ozone ($O_3$) oxidation of VOCs can produce peroxy

radical ($RO_2$), which reacts with $NO_x$ to produce ON (Perring et al., 2013), while the reaction between nitrate radicals ($NO_3$) and VOCs dominates the generation of ON in the nighttime (Rollins et al., 2009; Perring et al., 2013; Ng et al., 2017). Furthermore, the coexistence among OH, $O_3$ and $NO_3$ has been investigated in VOCs nocturnal oxidation (Brown and Stutz, 2012; Barber et al., 2018; Kwan et al., 2012; Chen et al., 2022). Compared with single oxidant, the introduction of multiple oxidants leads to the possible complex reaction mechanisms for VOCs. The regeneration of OH would change the $O_3$ oxidation process to form SOA (Sato et al., 2013). Chamber experiments show that SOA from $NO_3$ oxidation of VOCs are affected by oxidation of $NO_2$ by $O_3$ (Ng et al., 2017). Therefore, VOCs are oxidized through the synergistic effects of multiple oxidants, driving the chemical formation of ON. However, ON formation from the VOCs oxidation governed by mixing oxidants has not been fully understood. In particular, the impact of oxidation pathways on the ON formation and spatial distribution are still unclear.

As one of typical biogenic volatile organic compounds (BVOCs) (10% of monoterpenes), limonene is mostly emitted from citrus plants and coniferous trees, with a total emission rate of ~11 $Tg \cdot yr^{-1}$ (Guenther et al., 2012; Sindelarova et al., 2014). Limonene has a unique structure with an endocyclic double bond and an exocyclic double bond, which makes it reactive towards atmospheric oxidants (Surratt et al., 2008). Higher ON (30–72%) and SOA yields (17–40%) through $NO_3$-initiated oxidation of limonene than other monoterpenes have been observed in laboratory experiments (Fry et al., 2014; Hallquist et al., 1999; Spittler et al., 2006; Moldanova and Ljungström, 2000; Fry et al., 2011). It has been well demonstrated that limonene + $NO_3$ is most important pathway to form limonene-derived ON (Kilgour et al., 2024; Ehn et al., 2014; Jokinen et al., 2015; Zhao et al., 2015). Furthermore, recent study found the primary nitrooxy $RO_2$ formed through $NO_3$ addition to limonene occurs at both at endocyclic double bond and the exocyclic double bond. These products could undergo autoxidation, which is fast enough to $RO_2$ bimolecular reactions (Mayorga et al., 2022). The molecular compositions and formation mechanism of limonene-derived ON have been well investigated through observations and laboratory studies, while their description in models remains not explicit and advanced.

The early atmospheric model utilizes empirical yields and empirical coefficients for predicting limonene-derived SOA production in simulation (Yu et al., 2019). Currently, chemical mechanisms are simplified according to analogies with structurally similar compounds in most of regional and global

models due to simplicity and efficiency in calculation (Fisher et al., 2016; Li et al., 2023a). Nevertheless, previous model studies have not included the formation mechanism of limonene-derived ON in detail (Pye et al., 2015; Li et al., 2023a; Zare et al., 2019). Thus, incorporating explicit mechanisms is helpful to understand limonene-derived ON formation process and the influence of interaction between multiple oxidation pathways on ON formation.

Herein, we investigated the impacts of multiple oxidation pathways on limonene-derived ON using both chemical box model and global model, which were developed to include explicit chemical mechanisms for limonene-derived ON formation. The effect of competition among individual oxidation pathways on limonene-derived ON formation were discussed using a chemical box model based on proposed mechanisms. The simulation framework of explicit chemical mechanisms was integrated into global model to evaluate the spatial distributions of limonene-derived ON and contributions of individual oxidation pathways. This study presents a numerical simulation framework for atmospheric chemical processes and aims at enhancing the ability of models to simulate ON and understand the competition effects among atmospheric oxidation pathways on SOA formation, improving atmospheric composition forecasts and informing interaction between biogenic and anthropogenic emissions.

**2 Methodology**

**2.1 Limonene-derived ON formation mechanism**

In order to simulate ON via the gas-phase oxidation of limonene, the chemical mechanism used in our model was updated with gas-phase chemical mechanisms of limonene-derived ON based on recent laboratory studies (Mayorga et al., 2022) and Master Chemical Mechanism (MCM, v3.3.1). The explicit chemical mechanism of limonene-derived ON involves three initial oxidation pathways: OH-, $O_3$- and $NO_3$-initiated oxidation (Fig. 1). The detailed formulas of species could be found in Table S1. The updated explicit formation mechanisms were list in Fig. S1 and Table S2. Compared to the MCM mechanism, the chemical mechanism of limonene-derived ON formation used in this study is developed to include: (1) $NO_3$ addition at three different carbonsites. Based on previous laboratory studies, the exocyclic double bond oxidation branching ratio is ~0.03 (Fry et al., 2011; Donahue et al., 2007), while the branching ratios of the two endocyclic $C_{10}H_{16}NO_5$-$RO_2$ isomers are 0.65:0.35 (Mayorga et al., 2022). Thus, these branching ratios of the three $C_{10}H_{16}NO_5$-$RO_2$ isomers were used in our work. (2) Sequential

NO$_3$ oxidation reactions to form ON for all the products that contain double bonds from OH- and O$_3$-
initiated oxidation in MCM. The rate constants were set to be the same as those used in MCM for
limononaldehyde. (3) The formation of a ring-opened nitrooxy RO$_2$ in the presence of O$_2$ due to bond
scission of the two endocyclic nitrooxy RO, and its branching ratio was estimated (Draper et al., 2019;
Kurten et al., 2017; Guo et al., 2022). (4) H-shifts of the exocyclic C$_{10}$H$_{16}$NO$_4$-RO. (5) Bimolecular and
unimolecular reactions of the C$_{10}$H$_{16}$NO$_6$-RO$_2$ and C$_{10}$H$_{16}$NO$_7$-RO$_2$. The rate constants for the
bimolecular reactions are the same as those used in MCM, and autoxidation rate constants are calculated
by quantum chemical calculations (Mayorga et al., 2022). In addition, photolysis, widely recognized as
the predominant removal pathway of limonene-derived ON (Picquet-Varrault et al., 2020; Wang et al.,
2023), is included in our mechanism. While heterogeneous processes and hydrolysis of limonene-derived
ON are not included in our model, potentially resulting in a slight overestimation of simulated limonene-
derived ON concentrations, their contributions to ON removal are expected to be substantially smaller
than that of photolysis. Consequently, this omission introduces only minor uncertainties in our results.
In our explicit chemical mechanism, more intermediates and chemical processes of limonene-derived
ON were distinguished than simplified mechanisms used in previous models.

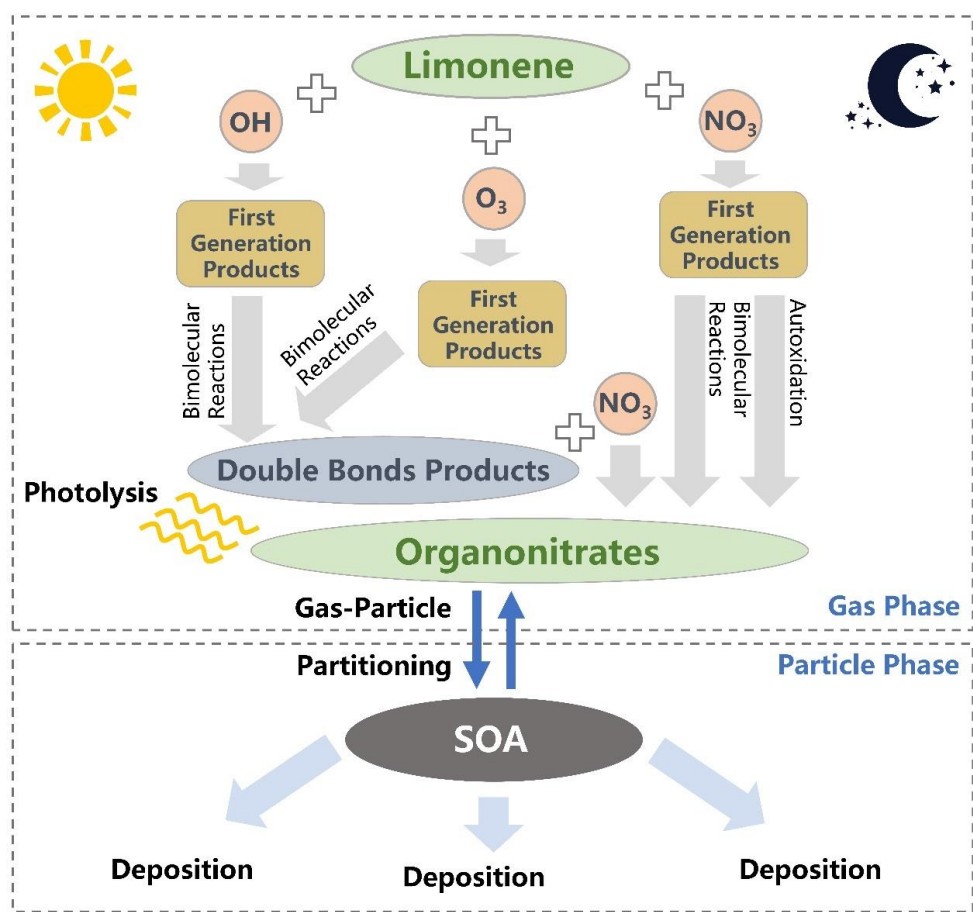


**Figure 1.** Schematic diagram of the limonene-derived ON formation pathways included in this work.
We assumed highly oxidated ON products ($C_{10}H_{13}NO_7$, $C_{10}H_{15}NO_4$, $C_{10}H_{15}NO_5$, $C_{10}H_{15}NO_6$,
$C_{10}H_{15}NO_7$, $C_{10}H_{15}NO_8$, $C_{10}H_{17}NO_5$, $C_{10}H_{17}NO_6$, $C_{10}H_{17}NO_7$) to be semi- to low-volatile which can
condense into the particulate phase upon formation. Their vapor pressures are estimated to calculate gas-
particle partitioning (Table S3).
The vapor pressures of the above-mentioned ON species were estimated using two group contribution
methods: EVAPORATION (Compernolle et al., 2011) and SIMPOL (Pankow and Asher, 2008). They
are both widely used structure activity relationship (SAR)-based group contribution models to predict
molecular vapor pressures. The key difference is that EVAPORATION considers proximity-based
functional group interactions, so it considers differences in the locations of functional groups, while
predictions from SIMPOL do not vary based on functional group locations. As a result, isomeric
compounds with the same functional groups but different structures may have different predicted vapor
pressures using EVAPORATION but the same using SIMPOL. Therefore, the EVAPORATION model
is preferred when chemicals structures are known while the SIMPOL model could be biased. In a recent
study, we showed that the EVAPORATION-based kinetic model predicts isoprene SOA more accurately
than the SIMPOL-based model, which underpredicts by ~ 20% (Shen et al., 2024).
In this work, because the chemical structures of the major ON species are known based on our recent
work (Mayorga et al., 2022), we adopted the EVAPORATION method in all our simulations. As the
EVAPORATION model input, the structures of the ON species from Mayorga et al. (2022) were
converted to SMILES strings. To illustrate the difference between the two models, the EVAPORATION-
predicted vapor pressures were compared with SIMPOL predictions (Table S3). The two methods predict
vapor pressures within one order of magnitude in most cases, which is typically considered acceptable
uncertainties for group contribution vapor pressure estimations.
**2.2 Chemical box model**
A zero-dimensional (0-D) chemical box model was used to examine the chemical processes of limonene-
derived ON, investigating the contributions of atmospheric oxidants and oxidant pathways. The chemical
mechanism presented in Fig. S1 and Table S2 was applied in this box model. To calculate the total
production of limonene-derived ON, processes such as photolysis, dilution, and deposition were ignored
for all chemical species in the model. The temperature was set to 298 K in the model. The initial
concentrations of limonene and other atmospheric components for all cases were set as shown in Table
S5. Limonene at a concentration of $1.0 \times 10^{11}$ molecules·cm$^{-3}$ was used as the precursor for ON, which
falls within the range of values reported in laboratory and observation studies (Guo et al., 2022; Luo et
al., 2023; Ham et al., 2016). The initial concentration of OH, O$_3$ and NO$_3$ spanned $1.0 \times 10^5$ to $1.0 \times 10^{19}$
molecules·cm$^{-3}$, $1.0 \times 10^{11}$ to $1.0 \times 10^{18}$ molecules·cm$^{-3}$ and $1.0 \times 10^9$ to $1.0 \times 10^{17}$ molecules·cm$^{-3}$,
respectively. The low values represent typical atmospheric concentrations of these species, which are
within the range of those reported in previous studies (Shen et al., 2021; Liu et al., 2023; Matsunaga and
Ziemann, 2019). The medium to high values represent extreme conditions, in order to investigate the
significant impact of oxidants on limonene-derived ON across a broad spectrum of oxidant levels.
Chamber experiments were simulated by the box model under ideal situation, which has been specifically
design to analyze chemical processes, while simulations under real atmospheric condition were carried
using global model in sect. 2.3. We conducted sensitivity tests (Sect. S1 in the supplement) to examine
oxidation pathways for formation of limonene-derived ON. Sensitivity tests under single initial oxidation
were set. Building upon this foundation, sensitivity tests with multiple oxidation pathways were
implemented: (1) introducing secondary oxidant across three concentration gradients under fixed primary
oxidant levels, followed by (2) increasing concentration of third oxidant with three concentration
gradients. A summary of all cases can be found in Table S4.
**2.3 Simulation of global limonene-derived ON**
We used the Community Earth System Model (CESM) version 1.2.2.1 coupled with the University of
Michigan Integrated Massively Parallel Atmospheric Chemical Transport (IMPACT) aerosol model with
a resolution of $1.9° \times 2.5°$ for this study. The CESM/IMPACT model have included a fully explicit gas-
phase photochemical mechanism to predict the formation of semi-volatile organic compounds (SVOCs)
which then partition to an aerosol phase (Lin et al., 2014), facilitating the incorporation of explicit
limonene-derived ON mechanism to simulate their global burden. The IMPACT aerosol module gets the
meteorology field from the CESM model at each time step, while changes in the aerosols in IMPACT
do not provide feedback to the CESM. The emission of precursors BVOCs are estimated by the Model
of Emissions of Gases and Aerosols from Nature inventory (MEGAN) coupled to CESM/IMPACT
model. The developed explicit gas phase chemical mechanism same as used in above chemical box model
was applied to simulate the formation of limonene-derived ON. The highly oxidated limonene-derived
ON considered as semi-volatile species partitioning from gas phase to particle phase contributes to SOA.
A base case (Case0) was designed to simulate limonene-derived ON under all three initial oxidation
pathways, and six sensitivity experiments were designed for simulating global burden limonene-derived
ON under two initial oxidation pathways (Case 1-3) and single initial oxidation pathway (Case 4-6),
respectively. Above seven cases were summarized in Supplementary Sect. S2 and Table S6.
**3 Results and discussion**
**3.1 Limonene-derived ON formation through individual initial oxidation pathway.**
We employed a chemical box model to simulate limonene-derived ON formed through three initial
oxidation pathways, considering various oxidant concentrations (Fig. 2). These simulations were
designed to evaluate the effect of increasing oxidant concentrations on the yield of limonene-derived ON
from each initial oxidation pathway. In the case with individual OH oxidation pathway, the concentration
of limonene-derived ON increases as the initial OH concentration increases (Fig. 2a), following a pattern
to that of limonene consumption (Fig. S2a). Initial OH concentration increases from $1.0 \times 10^5$ to $1.0 \times 10^{19}$
molecules·$cm^{-3}$, resulting in ~20.0-fold increase in the production of limonene-derived ON. At this stage,
limonene is not completely consumed by OH, indicating that higher initial OH concentration will
increase consumption of limonene to produce more ON.

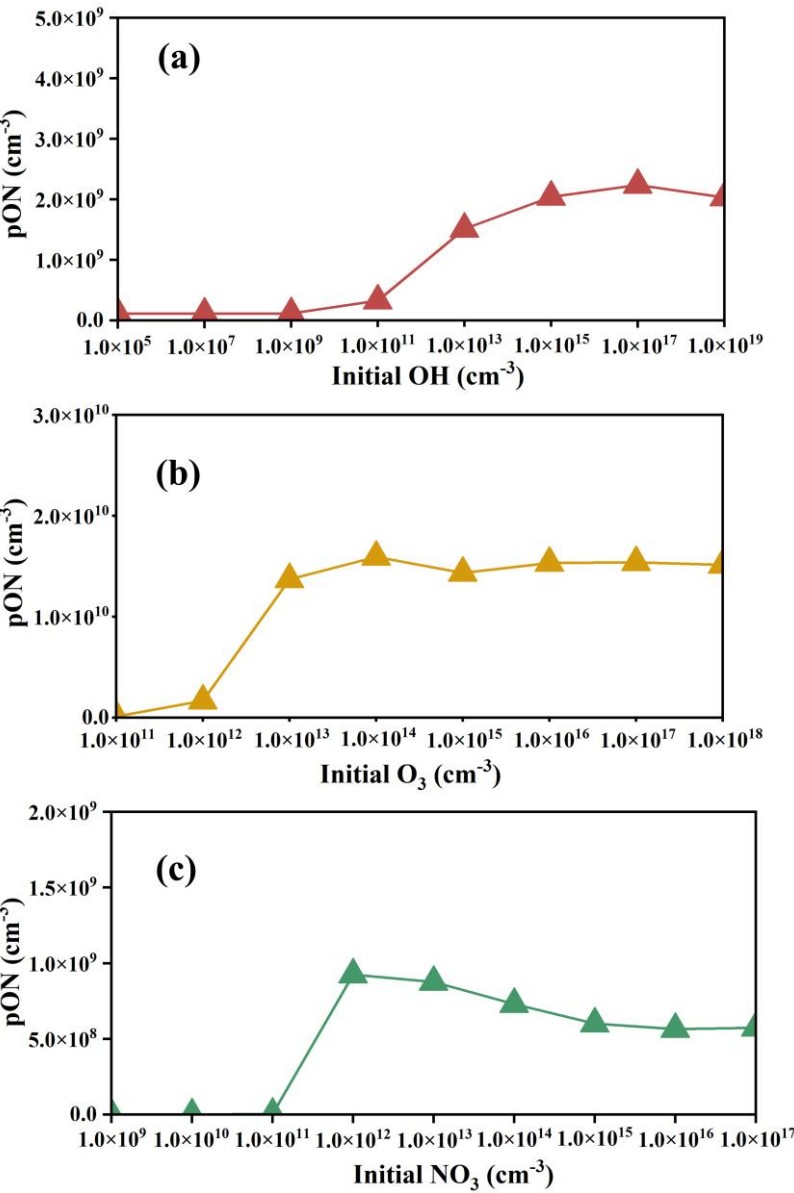


**Figure 2.** Variations of limonene-derived ON in individual oxidation pathway under different oxidant concentrations.
The triangles represent concentration of limonene-derived ON in each experiment. The lines represent the trend of
limonene-derived ON. The three datapoint colors represent three initial oxidation pathways (red for OH-initiated
oxidation, yellow for $O_3$-initiated oxidation, green for $NO_3$-initiated oxidation).

In the case with individual $O_3$ oxidation pathway, the limonene-derived ON increases first and then
maintains a relatively stable production with the increase of initial $O_3$ concentration (Fig. 2b). Limonene
is not completely consumed when initial $O_3$ concentrations below $1.0 \times 10^{14}$ molecules·$cm^{-3}$. Increased
consumption of limonene lead to an increase in ON production with increased $O_3$ concentration.
Same to the cases of the OH- and $O_3$-initiated oxidation pathways, limonene-derived ON increases
when initial $NO_3$ concentrations below $1.0 \times 10^{12}$ molecules·$cm^{-3}$ could be caused by incompletely
consumed limonene (Fig. 2c). The increased consumption of limonene with increase in concentrations
of $NO_3$ lead to the increased production of ON. However, different from the cases of OH- and $O_3$-initiated
oxidation pathways, as initial $NO_3$ concentrations continued to increase, limonene-derived ON
production decrease. When limonene-derived ON concentrations reached steady state within 30 minutes,
compared to the case with initial $NO_3$ concentration of $1.0 \times 10^{12}$ molecules $cm^{-3}$, reaction of LIMAL and
$NO_3$ become the dominant pathway in the case with initial $NO_3$ concentration of $1.0 \times 10^{17}$ molecules·$cm^{-}$
$^3$. The lower yield of the $NO_3$ oxidation pathway (9.2%) of LIMAL relative to OH oxidation pathway
(28.8%) results in decreased limonene-derived ON (green box in Fig. S1). The results mean that at low
initial oxidant concentration, limonene-derived ON shows a strong dependence on initial oxidant
concentration, and the dependence on intermediate reaction rates becomes more important at high initial
oxidant concentration.
In addition, average concentration of ON of OH-, $O_3$- and $NO_3$-initiated oxidation pathways when
oxidations are sufficient are calculated separately. The $O_3$-initiated oxidation pathway ($1.5 \times 10^{10}$
molecules·$cm^{-3}$ limonene-derived ON produced) yields more ON than the OH- ($2.1 \times 10^9$ molecules·$cm^{-}$
$^3$ limonene-derived ON produced) and $NO_3$-initiated ($7.1 \times 10^8$ molecules·$cm^{-3}$ limonene-derived ON
produced) oxidation pathways when limonene initial concentration is constant. This indicates that under
initial conditions with sufficient oxidation, $O_3$-initiated oxidation pathway of limonene has highest yield
of ON, which is about 15.0%, while that is low by OH- (2.1%) and $NO_3$-initiated (0.7%) oxidation
pathway. This difference in the ON yield among various oxidation pathways will be used to explain the
contributions of each oxidation pathway to ON concentration in the following discussion.

**3.2 Effects of multiple oxidation pathways on limonene-derived ON formation.**

Compared to the simulation scheme with individual oxidation pathway discussed above, introducing multiple oxidation pathways leads to comprehensive competition among them, which results in a nonlinear response of ON concentration to changes in the initial concentrations of oxidants. Figure 3 shows the dependence of limonene-derived ON on initial concentration of oxidants when include two initial oxidation pathways. The addition of oxidants has various effects on the yield of limonene-derived ON. When the initial concentration of oxidant is low ($1.0\times10^5$ molecules·cm$^{-3}$ for OH, $1.0\times10^{11}$ molecules·cm$^{-3}$ for $O_3$, $1.0\times10^9$ molecules·cm$^{-3}$ for $NO_3$), the initial limonene will not be completely consumed. In all case with low concentration of oxidants, adding another oxidant with oxidation pathway will increase consumption of limonene, leading to increase in the limonene-derived ON production. When the initial concentration of oxidants is high, limonene will be nearly or completely consumed. In these cases, the production of limonene-derived ON will be determined by the competition between the two oxidation pathways. The product of limonene-derived ON steadily increased as the initial concentration of $O_3$ increases from 0 to $1.0\times10^{18}$ molecules·cm$^{-3}$ when the initial concentration of OH or $NO_3$ is constant (Fig. 3a, f). According to the chemical mechanism applied in the model, the reaction between limonene and $O_3$ has higher rate than OH and $NO_3$ in these cases (Table S7). As a result, in the presence of $O_3$, the oxidation of limonene with $O_3$ proceeds more rapidly than with OH or $NO_3$, leading to higher concentration of limonene-derived ON due to the high yield of $O_3$ oxidation pathway as discussed in above section (compare Fig. 2b with Fig. 2a and 2c). In contrast, compared to only including $O_3$ oxidation pathway, adding oxidation pathways with OH or $NO_3$ will result in a decrease in limonene-derived ON production (Fig. 3c, d), because some limonene that would have reacted with $O_3$ is instead converted to ON through the OH or $NO_3$ pathways with lower yield. Therefore, the dominant oxidation pathway and its ON yield determine the impact of the competition between the two oxidation pathways on the final limonene-derived ON production. A similar phenomenon observed in laboratory study shows that $NO_x$ influences γ-terpinene ozonolysis by enhancing $NO_3$ production at high $NO_x$ levels, which subsequently leads to $NO_3$ preferentially consuming γ-terpinene over $O_3$ (Xu et al., 2020), illustrating the competition between oxidants. The addition of the OH-initiated oxidation pathway results in a small increase in ON production compared to $NO_3$-initiated oxidation alone (Fig. 3e), due to the slightly higher yields of limonene-derived ON for OH-initiated oxidation pathway. The ON production would not

change much when add the NO$_3$-initiated oxidation pathway compared to the case with OH-initiated
oxidation pathway alone (Fig. 3b) because of unchanged the main initial oxidation pathway. These
sensitivity experiments suggest that competition of oxidation pathways plays an important role in
formation of limonene-derived ON.

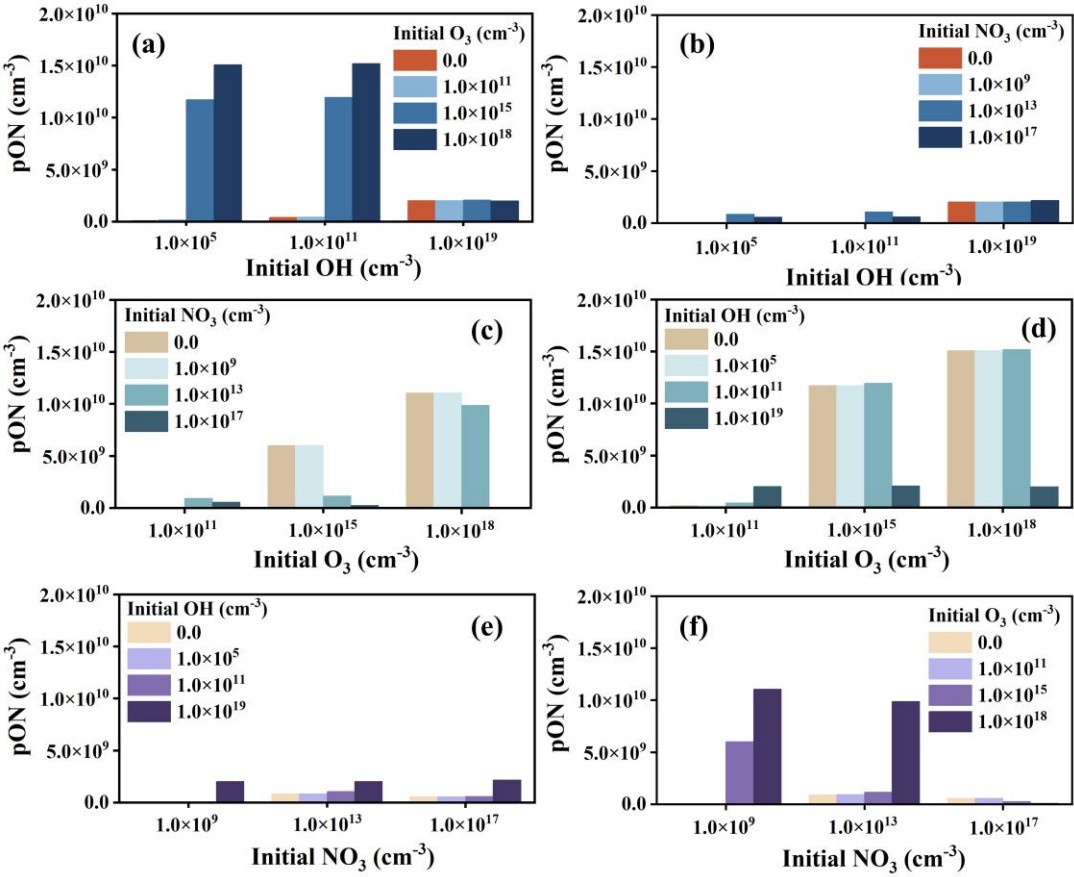


**Figure 3.** Simulated limonene-derived ON in two initial oxidation pathways under different oxidant conditions,
including variation of production of limonene-derived ON with adding (a) initial O$_3$ concentration and (b) initial
NO$_3$ concentration in the three OH levels; variation of limonene-derived ON production with adding (c) initial OH
concentration and (d) initial NO$_3$ concentration in the three O$_3$ levels; variation of limonene-derived ON production
with adding (e) initial OH concentration and (f) initial O$_3$ concentration in the three NO$_3$ levels.
Based on the production variations of limonene-derived ON in the cases with one and two initial
oxidation pathways discussed above, the comprehensive impact of multiple oxidants on limonene-
derived ON formation in the cases with multiple initial oxidation pathways are analyzed (Fig. 4). The
results can be summarized into three types. The Type 1 is the cases when limonene is not completely
consumed (Fig. S4). When two initial oxidant concentration is low (Fig. 4a, d, g) and medium (Fig. 4d,
g), the addition of third oxidant increases the production of limonene-derived ON because the addition

of the third oxidant increases consumption of limonene. If the oxidant concentration is sufficient to consume up limonene, the production of limonene-derived ON will be determined by the competition between initial oxidation pathways. The Type 2 is the cases with large changes of limonene-derived ON. Under low $NO_3$ and moderate and high $O_3$ conditions, the production of limonene-derived ON decreases with adding OH (Fig. 4a, b), because some limonene that would have reacted with $O_3$ is instead converted to ON through the OH pathways with lower yield. The formation of limonene-derived ON shows similar pattern for Type 2 in Figure 4h and i. On the one hand, the yield of limonene-derived ON from $NO_3$-initiated oxidation is lowest, so the production of limonene-derived ON will decrease when the formation of limonene-derived ON from this pathway becomes the dominant formation route. On the other hand, adding $NO_3$-initiated oxidation pathway also consumes $NO_3$ that would have reacted with the product of the OH- and $O_3$-initiated oxidation, resulting in decrease production of limonene-derived ON. The changes in ON production with constant initial concentration of limonene and various oxidation pathways indicate the interactions of different oxidation process of limonene. In contrast to OH- and $NO_3$-initiated oxidation pathway, adding oxidation pathways with $O_3$ will result in increase in limonene-derived ON production (Fig. 4e), due to higher yield of limonene-derived ON from $O_3$-initated oxidation pathway than OH- and $NO_3$-initated oxidation pathways. Since the yield of limonene-derived ON of OH-initiated oxidation is higher than $NO_3$-initiated oxidation, the production of limonene-derived ON decreases (Fig. 4c) as the main oxidation pathway changes from $NO_3$ to OH oxidation (Table S8). Additionally, in some sensitivity experiments (Fig. 4d-i), ON concentration do not change much with the addition of an oxidation pathway (Type 3). This could be explained by minimal competition with the rapid main oxidation pathway. These sensitivity experiments suggest that the limonene-derived ON production in the simulated system are not only controlled by limonene concentration, but also affected by synergic effect of multiple oxidants and oxidation pathways.

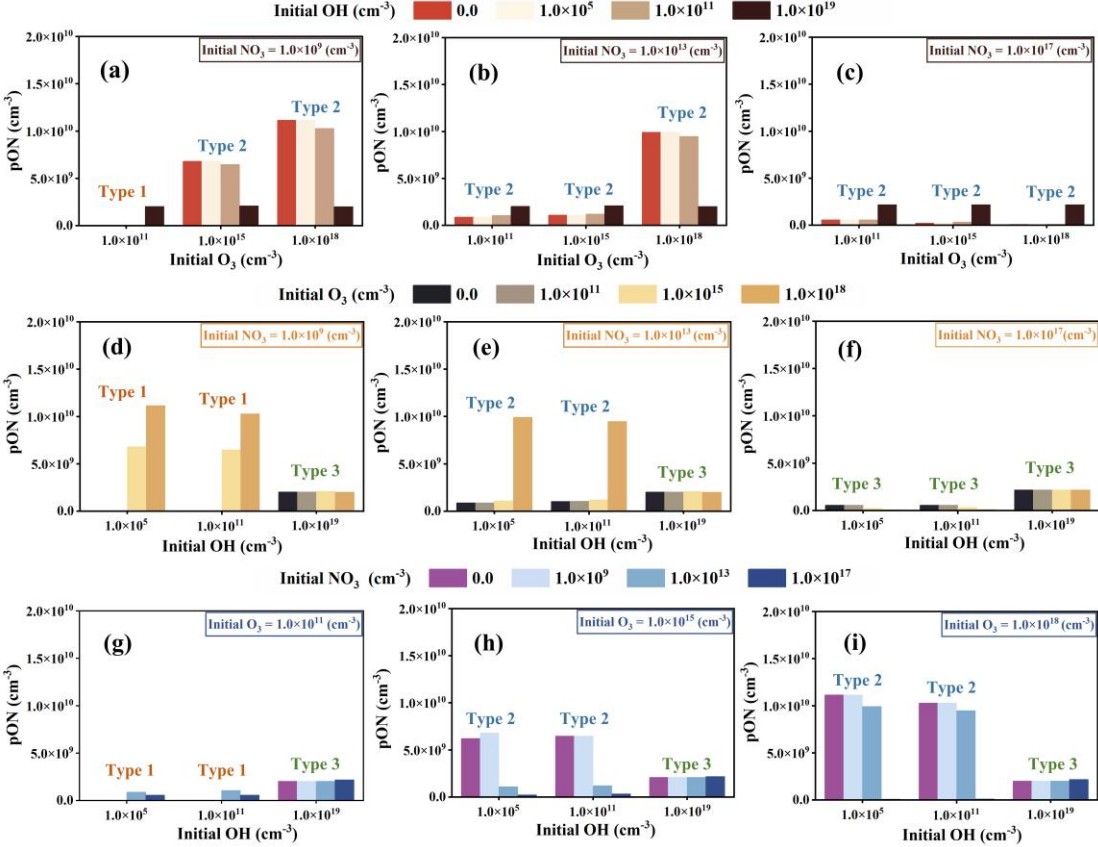

**Figure 4.** The influence of adding OH-, O₃- and NO₃-initiated oxidation pathways on the production of limonene-derived ON under different oxidant conditions, including variation of limonene-derived ON production with adding initial OH concentration in the three O₃ levels under (a) low, (b) moderate and (c) high NO₃ levels; variation of limonene-derived ON production with adding initial O₃ concentration in the three OH levels under (d) low, (e) moderate and (f) high NO₃ levels; variation of limonene-derived ON production with adding initial NO₃ concentration in the three OH levels under (d) low, (e) moderate and (f) high O₃ levels. In each panel, the types marked on the columns show the cases when limonene is not completely consumed (type 1) and almost completely consumed (large (type 2) and small (type 3) changes in limonene-derived ON production).

**3.3 Contribution of each oxidation pathway to global limonene-derived ON.**

Global simulation using CESM/IMPACT model was performed to characterize the spatial and temporal distribution of limonene-derived ON and the contributions of each oxidation pathway to global burden. Incorporation of formation of limonene-derived ON reduces underestimation of simulated ON by comparison with observations summarized in the literature (Sect. S3 in the supplement) (Li et al., 2023b). The spatial distribution of limonene-derived ON is shown in Fig. 5a. The simulated global mean limonene-derived ON burden is about 60.5 μg·m⁻², and the highest burdens (>500 μg·m⁻²) are predicted over tropical forest regions of central Africa. As the primary precursor of limonene-derived ON, the concentration of limonene dominates the yield of these ON compounds. The seasonal cycle of simulated limonene-derived ON is presented in Fig. S6, which is mainly depend on limonene levels. Global average

limonene-derived ON burden peaks in the summer (69.2 $\mu g \cdot m^{-2}$) due to highest global average limonene
concentration (Fig. S7b), while the large burden of limonene-derived ON in fall is driven by the presence
of both high limonene concentration (Fig. S7c) and NO concentration (Fig. S8c) compared to spring and
winter. In contrast, the burden of limonene-derived ON is lowest in winter (48.1 $\mu g \cdot m^{-2}$) because of
lowest concentration of limonene (Fig. S7d). Beyond the effects of limonene and NO concentrations,
oxidant levels and oxidation pathways also affect the formation mechanisms and production of limonene-
derived ON, which may explain the highest burden in regions such as Central Africa, rather than Amazon
where limonene concentrations are highest over the world (Fig. S9a). The concentration of oxidants is
inherently low in Amazon (Fig. S9b-d) and oxidant scavenging in the presence of isoprene with high
concentrations greatly reduce the photochemical formation of limonene-derived ON (Mcfiggans et al.,
2019). Thus, oxidant competition with isoprene leads to low burden of limonene-derived ON in Amazon
despite the highest burden of limonene there. Therefore, high concentrations of limonene-derived ON
can only form when both high limonene and oxidant concentrations are present simultaneously.

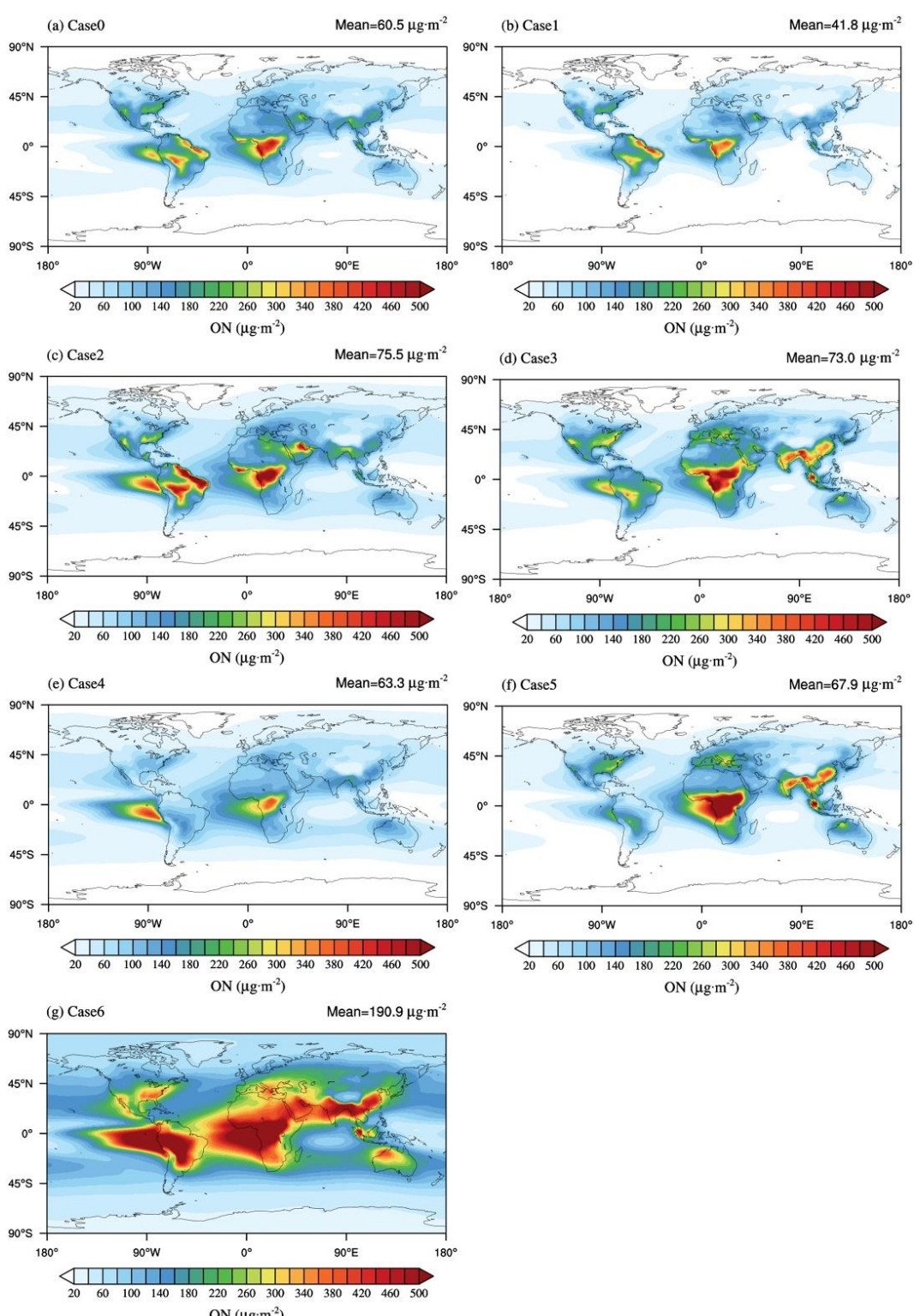

**Figure 5.** Annual mean column concentration of limonene-derived ON with different simulation schemes. (a) Run
with three initial oxidation pathways (Case0), (b) without OH-initiated oxidation pathway (Case1), (c) without $O_3$-
initiated oxidation pathway (Case2), (d) without $NO_3$-initiated oxidation pathway (Case3), (e) without $O_3$- and $NO_3$-
initiated oxidation pathways (Case4), (f) without OH- and $NO_3$-initiated oxidation pathways (Case5) and (g) without
OH- and $O_3$-initiated oxidation pathway (Case6).
To quantify the contribution of each oxidation pathway to the formation of limonene-derived ON in
different regions, we conducted a series of sensitivity experiments (Case1 to 6 introduced in method
section) on the oxidation pathways (Fig. 5b-g). Our simulations indicate that increasing $O_3$ and $NO_3$-
initiated oxidation pathways result in 15.5% and 18.0% increase of global average burden of limonene-
derived ON, respectively, compared to OH-initiated oxidation pathway alone (Fig. S10a, b). This is
primarily because higher yields of limonene-derived ON associated with the $O_3$- and $NO_3$-initiated
oxidation pathways compared to OH-initiated oxidation pathways. When compared to $O_3$-initiated
oxidation pathway alone (Fig. S10c, d), the addition of OH- or $NO_3$-initiated pathways result in increased
burden of limonene-derived ON in the limonene-sufficient region (e.g. Amazon), owing to adding a
limonene-derived ON formation pathway to consume more limonene. However, in the limonene-
deficient yet $NO_3$-sufficient regions (e.g. Central Africa, Mediterranean, and middle and low latitude of
Asia), increasing the OH- or $NO_3$-initiated oxidation pathways reduces the burden of limonene-derived
ON. This occurs because the oxidation of limonene by OH or $NO_3$ suppresses $O_3$-initiated oxidation,
which otherwise produces limonene-derived ON with a high yield. Additionally, if limonene undergoes
oxidation by $NO_3$, the availability of $NO_3$ for the nitration of OH- and $O_3$-initiated oxidation products of
limonene will decrease, resulting in a reduction in limonene-derived ON. The addition of OH- and $O_3$-
initiated oxidation pathways reduces global average burden of limonene-derived ON by 60.5% and 78.1%
respectively, compared to the case with $NO_3$-initiated oxidation pathway alone (Fig. S10e, f). This
reduction is likely due to insufficient $NO_3$ oxidation at night to further oxidize intermediates produced
from OH- and $O_3$-initiated limonene oxidation during the day, limiting the formation of limonene-derived
ON at night.
The burden of limonene-derived ON undergoes a noticeable change when an additional oxidation
pathway is introduced to the existing two pathways (Fig. S11). Adding OH-initiated oxidation pathway
increases the global average burden of limonene-derived ON from 41.8 to 60.5 $\mu g \cdot m^{-2}$, by 44.7%, while
adding $O_3$-initiated oxidation pathway decrease that from 75.5 to 60.5 $\mu g \cdot m^{-2}$, by 19.9% (Fig. S11a, b),
which was attributed to the competition between the OH and $O_3$ oxidation pathways for reactions with
limonene. Compared to the simplified condition in simulation using chemical box model, global
simulation considers diurnal variations of oxidation. When the $O_3$-initiated oxidation pathway produces
the same amount of limonene-derived ON as the OH-initiated pathway, it consumes more $NO_3$. As a
result, increasing the $O_3$ oxidation pathway reduces the availability of $NO_3$ for the nitration of
intermediate oxidation products in the nighttime, thereby lowering the total limonene-derived ON yield
across all three pathways. In contrast, enhancing the OH oxidation pathway increases the total yield.
Moreover, the addition of the $NO_3$-initiated oxidation pathway increases burden of limonene-derived ON
in the limonene-sufficient region even over 150 $\mu g \cdot m^{-2}$ (Fig. S11c). However, in the region with high
$NO_3$ concentration, the burden of limonene-derived ON decreases (Fig. S11c) because the $NO_3$-initiated
oxidation pathway yields less limonene-derived ON than the $O_3$- and OH-initiated oxidation pathways.
These results highlight the different nonlinear responses of limonene-derived ON to multiple oxidation
pathways under varying oxidation conditions and precursor concentrations. This discrepancy highlights
differences between global-scale dynamics and idealized box model conditions, emphasizing the
importance of developing explicit chemical mechanisms in global models for understanding SOA
formation processes. Prior laboratory study has also demonstrated that investigating the response of ON
reveals complex and nonlinear behaviour with implications that could inform expectations of changes to
ON concentrations as efforts are made to reduce oxidant concentrations (Mayhew et al., 2023).
**4 Conclusion and implications**
In this work, the explicit chemical mechanism is developed to simulate formation and spatial distribution
of limonene-derived ON using a chemical box model and global model CESM/IMPACT. Under multiple
initial oxidation pathways, limonene-derived ON shows non-linear variations with different oxidant
conditions, which is controlled by the synergetic effects of multiple oxidants. When limonene is not
consumed, adding another oxidant with oxidation pathway will increase limonene-derived ON due to
increased consumption of limonene. When limonene is completely consumed, limonene-derived ON
production is dominated by competition of oxidation pathways. The production of limonene-derived ON
is increased by $O_3$-initiated oxidation pathway while decreased by OH and $NO_3$-initiated oxidation
pathway. This is mainly because limonene oxidated by $O_3$ produces more ON than OH and $NO_3$, resulting
from the simulation under individual initial oxidation pathway.
The global model simulation indicates that oxidation process is important for limonene-derived ON
formation in addition to limonene concentration. Global limonene-derived ON burden decreases by 19.9%
and 17.3% due to $O_3$- and $NO_3$-initiated oxidation pathway, while OH-initiated oxidation pathway
increases global limonene-derived ON burden by 44.7% compared the case only including the other two
oxidation pathways. These differences can be attributed to the complex nonlinear response of limonene-
derived ON yield to different reaction pathways under varying precursor and oxidant conditions.

391        The chemical mechanism of ON formation may influence the formation and spatial distribution of

ON. We only include main oxidation process published to date in the model, while some pathways (i.e.
Heterogeneous $NO_3$ reactions) of ON is missing in this work. Gas-phase oxidation in our mechanism is
considered as the dominant formation pathway of ON (Fan et al., 2022; Perring et al., 2013). Future
inclusion of newly identified and quantified ON chemistry may lead to unpredictable nonlinear impacts
on simulation outcomes. Although uncertainties remain in simulating limonene-derived ON due to
limited knowledge of its formation mechanism, this work offers an improvement in the global model's
ability to simulate ON and presents a methodological framework for simulating SOA and their chemical
processes. This framework can be used in the future to improve SOA burden predictions and provide a
comprehensive understanding of the complex interactions between multiple oxidation pathways, which
are crucial for SOA formation (Chen et al., 2022; Zang et al., 2024). Quantitative understanding of these
complex interactions in contributing to SOA formation can definitely facilitate better understanding the
contributions of interactions and antagonistic actions between anthropogenic and natural emissions to
atmospheric aerosols. These works provide valuable insights for making more effective secondary
aerosol pollution control strategies to improve air quality.
**Data availability.** Simulation data are available upon request to the corresponding authors.
**Author contributions.** QG and JZ designed the study, developed the chemical box model and global
model conducted the simulations, analyzed the data, and wrote the manuscript. HZ and BL provided the
laboratory data. PF, LC, YS, HL, YL and YX contributed to the discussion and revision of the paper.
**Competing interests.** The authors declare no competing financial interest.
**Disclaimer.** Publisher's note: Copernicus Publications remains neutral with regard to jurisdictional
claims in published maps and institutional affiliations.

**Acknowledgments.** The authors acknowledge the financial support of the National Natural Science

Foundation of China.

**Financial support.** This study was supported by the National Natural Science Foundation of China

(Grant 42177082).

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
