# Peer review of "Competing multiple oxidation pathways shape"

_EGUsphere, 2025_

## Referee Comment (RC2)

**Review of "Competing multiple oxidation pathways shape atmospheric limonene-derived organonitrates simulated with updated explicit chemical mechanisms" by Guo et al.**

**General Comments**

This manuscript addresses a critical gap in atmospheric chemistry modeling by developing and implementing an explicit chemical mechanism for limonene-derived organonitrates (ON) in both box and global models. The incorporation of 90 gas-phase reactions and 39 intermediates represents a substantial advance over simplified schemes, and the sensitivity experiments vividly illustrate nonlinear interactions among OH, $O_3$, and $NO_3$ oxidation pathways. The explicit chemical mechanisms developed here significantly advance the field and offer a robust framework for future studies on secondary organic aerosols. The work is timely, given the increasing recognition of ON's role in secondary organic aerosol formation. I support publication after minor revisions to improve clarity in following comments.

**Specific Comments**

1. The introduction effectively contextualizes the importance of ON in SOA and the gaps in current understanding. However, the transition from general SOA/ON to limonene-specific mechanisms could be smoother. Consider briefly mentioning the structural uniqueness of limonene earlier (e.g., around Line 59) to better justify its selection as the focus of this study.

2. Lines 73-80: The discussion of model limitations is useful, but it would be helpful to explicitly state how this study addresses these limitations (e.g., by incorporating explicit mechanisms). This could be clarified further.

3. Lines 118: The vapor pressure estimation methods are well-explained, but a brief discussion on the potential uncertainties or limitations of these methods (e.g., sensitivity to molecular structure) would strengthen this section.

4. Lines 147-149: The global model setup is clearly described, but it would be helpful to briefly justify the choice of CESM/IMPACT over other models, especially given the focus on explicit mechanisms.

5. The decrease in ON production at high $NO_3$ concentrations (Line 185) is attributed to the dominance of the LIMAL + $NO_3$ pathway (yield: 9.2%). The abrupt transition in Figure 2c (from increase to decrease) warrants a brief discussion of the timescales involved. Is this a kinetic effect (e.g., $NO_3$ outcompeting other pathways) or a thermodynamic limitation?

6. Figure 2: The trends in ON production under different oxidant concentrations are clearly presented. However, the discussion of the $NO_3$-initiated pathway (Lines 183-191) could benefit from a more explicit comparison to the OH- and $O_3$-initiated pathways to highlight the mechanistic differences.

7. Figure 2: The y-axis label should specify whether ON concentrations are gas-phase, particlephase, or total?

8. The explanation for low ON burdens in the Amazon (despite high limonene) due to oxidant competition with isoprene (Lines 298-305) is plausible but speculative without quantification. Consider adding a sentence referencing modeled oxidant budgets or prior studies showing isoprene's oxidant sink role

9. The 44.7% increase in ON burden from adding OH (Line 330) contrasts sharply with the box model's lower OH-initiated yield (2.1%, Line 195). This discrepancy should be explicitly addressed: Is it driven by regional OH abundance (e.g., tropical OH hotspots) or nonlinear interactions in the global model?

10. Lines 341-345: The nonlinear responses to multiple pathways are well-explained, but a brief mention of how these findings align with or diverge from prior laboratory or modeling studies would provide broader context.

11. Lines 364-365: A specific example of a missing mechanism or future experimental validation could make this more concrete. Are there missing pathways (e.g., heterogeneous $NO_3$ reactions) that could alter conclusions?

12. The implications for policy or air quality management could be expanded slightly, given the anthropogenic-biogenic interaction focus.

**Technical Corrections**
1. Line 47: "evaded" should likely be "avoided".
2. Line 104: "limonaldehyde" → "limononaldehyde" (consistency with MCM).
3. Line 132: "$1.0 \times 10^{11}$ molecules·cm^-3" seems high for limonene; consider clarifying if this is a typo or based on specific experimental conditions.
4. Line 224: "phenomena" should be "phenomenon".

---

## Author Comment (AC1)

**Response to the reviewers' comments on "*Competing multiple oxidation pathways shape atmospheric limonene-derived organonitrates simulated with updated explicit chemical mechanisms*"**

Qinghao Guo[1], Haofei Zhang[2], Bo Long[3], Lehui Cui[1], Yiyang Sun[1], Hao Liu[1], Yaxin Liu[1], Yunting Xiao[1], Pingqing Fu[1] and Jialei Zhu[1,*]

[1] Institute of Surface-Earth System Science, School of Earth System Science, Tianjin University, Tianjin, 300072, China;

[2] Department of Chemistry, University of California, Riverside, California 92521, USA;

[3] College of Materials Science and Engineering, Guizhou Minzu University, Guiyang 550025, China.

* Corresponding author.

 Email address: zhujialei@tju.edu.cn

Dear Editors and Reviewers:

Thanks for your letter and for the valuable comments. We carefully studied all comments and revised the manuscript accordingly. The updates in the manuscript are marked in red, which is quoted in *blue italics* in this response letter. The main revisions in the paper and replies to comments point by point are as following.

In response to the reviewers' comments, we have made the following key revisions: 1. We clarified previously ambiguous statements to improve the clarity of introduction section. 2. We included detailed information for our updated chemical mechanism and calculation of vapor pressure. 3. We provided detailed explanations of nonlinear change of limonene-derived ON by multiple oxidation pathways in global simulation and expanded the implications.

**Response to the comments from reviewer 1**

**Summary:**

This study presents a comprehensive investigation into the formation of limonene-derived organonitrates (ON) under competing oxidation pathways ($O_3$, OH, $NO_3$) using both a chemical box model and a global model. The work addresses a critical gap in understanding how multiple oxidation pathways influence ON formation and their spatiotemporal variability, offering valuable insights into secondary organic aerosol (SOA) dynamics. The integration of explicit chemical mechanisms into global models is a significant advancement, enhancing predictive capabilities for atmospheric chemistry. The findings are novel and relevant to air quality and climate change, which would be interesting to the readers of ACP and the community. However, several aspects require clarification or improvement to strengthen the scientific rigor and clarity of the manuscript.

Response to **Overall issues**:

We thank your appreciation to our paper and your helpful major and minor comments. Replies to your questions here were listed one by one below.

**Major comments:**

1. **The branching ratios and rate constants for NO₃ addition (e.g., 0.63:0.34:0.03) and autoxidation pathways are critical to model outcomes. However, the justification for these values (e.g., experimental validation vs. analogy with similar compounds) is insufficiently detailed.**

**Response:**

Thank you for your suggestion. Based on your comments, we have added detailed information of branching ratios and rate constants for NO₃ addition and autoxidation pathways based previous laboratory studies with references.

We have updated in the manuscript:

"*Based on previous laboratory studies, the exocyclic double bond oxidation branching ratio is ~0.03 (Fry et al., 2011; Donahue et al., 2007), while the branching ratios of the two endocyclic $C_{10}H_{16}NO_5$-$RO_2$ isomers are 0.65:0.35 (Mayorga et al., 2022). Thus, these branching ratios of the three $C_{10}H_{16}NO_5$-$RO_2$ isomers were used in our work.*" from Line 102 to Line 106.

"*The rate constants for the bimolecular reactions are the same as those used in MCM, and autoxidation rate constants are calculated by quantum chemical calculations (Mayorga et al., 2022).*" from Line 112 to Line 113.

References:
Fry, J. L., Kiendler-Scharr, A., Rollins, A. W., Brauers, T., Brown, S. S., Dorn, H. P., Dubé, W. P., Fuchs, H., Mensah, A., Rohrer, F., Tillmann, R., Wahner, A., Wooldridge, P. J., and Cohen, R. C.: SOA from Limonene: Role of NO3 in Its Generation and Degradation, Atmos. Chem. Phys., 11, 3879-3894, https://doi.org/10.5194/acp-11-3879-2011, 2011.

Donahue, N. M., Tischuk, J. E., Marquis, B. J., and Huff Hartz, K. E.: Secondary organic aerosol from limona ketone: insights into terpene ozonolysis via synthesis of key intermediates, Phys Chem Chem Phys, 9, 2991-2998, https://doi.org/10.1039/b701333g, 2007.

Mayorga, R., Xia, Y., Zhao, Z., Long, B., and Zhang, H.: Peroxy Radical Autoxidation and Sequential Oxidation in Organic Nitrate Formation during Limonene Nighttime Oxidation, Environ. Sci. Technol., 15337-15346,

https://doi.org/10.1021/acs.est.2c04030, 2022.

**2. While the comparison with observations is mentioned (Sect. S3), the manuscript lacks quantitative validation metrics (e.g., correlation coefficients, normalized mean bias) for limonene-derived ON in the global model.**

**Response:**

We appreciate your recommendations about the comparison. We have added quantitative validation metrics for limonene-derived ON in the global model. Please see Line 133-138 in supplement "*Simulations excluding limonene-derived ON formation largely underestimated the results by 92% (Table S10), which may be due to incomplete consideration of the precursors and formation process of ON in the mechanism. The incorporation of the limonene-derived ON formation mechanism improved global simulation of ON, resulting in increased simulated concentration of ON (0.05-0.50 μg·m$^{-3}$ to achieve 85.2% lower than observations, especially at forest and coastal sites.*"

We also added a table in supplement to quantify simulation performance before and after model refinement.

**Table S10.** The NMB between observation and simulation and in different schemes.

| Scheme | Observation (ng·m$^{-3}$) | Simulation (ng·m$^{-3}$) | NMB[a] | Description |
|--------|---------------------------|--------------------------|--------|-------------|
| 1 | 1296.2 | 90.2 | -93.0% | Exclusion of the limonene-derived ON formation |
| 2 | 1296.2 | 192.2 | -85.2% | Incorporation of the limonene-derived ON formation |

[a]NMB = normalized mean bias

**3. The statement that "enhancing OH-initiated pathways increases ON burden by 44.7%" (Sect. 3.3) contrasts sharply with box model results showing lower**

**OH-initiated yields. The explanation for this discrepancy (e.g., regional precursor availability, diurnal variations) is underdeveloped.**

**Response:**

Thank you for the comments. We have explained the discrepancy between the chemical box model and global model in their simulated impact of the OH-initiated oxidation pathway on limonene-derived organonitrates.

We have updated in the manuscript:

"*Compared to the simplified condition in simulation using chemical box model, global simulation considers diurnal variations of oxidation. When the $O_3$-initiated oxidation pathway produces the same amount of limonene-derived ON as the OH-initiated pathway, it consumes more $NO_3$. As a result, increasing the $O_3$ oxidation pathway reduces the availability of $NO_3$ for the nitration of intermediate oxidation products in the nighttime, thereby lowering the total limonene-derived ON yield across all three pathways.*" in Line 365 to Line 370.

4. **The exclusion of heterogeneous reactions or aerosol-phase processes (e.g., hydrolysis of ON) may underestimate ON loss pathways. You should show this uncertainty.**

**Response:**

Thank you for the comments. We have added a statement regarding the removal pathways for limonene-derived organonitrates within the chemical mechanism. We have updated in the manuscript:

"*In addition, photolysis, widely recognized as the predominant removal pathway of limonene-derived ON (Picquet-Varrault et al., 2020; Wang et al., 2023), is included in our mechanism. While heterogeneous processes and hydrolysis of limonene-derived ON are not included in our model, potentially resulting in a slight overestimation of simulated limonene-derived ON concentrations, their contributions to ON removal are*

*expected to be substantially smaller than that of photolysis. Consequently, this omission introduces only minor uncertainties in our results.*" in Line 113 to Line 119.

Reference

Picquet-Varrault, B., Suarez-Bertoa, R., Duncianu, M., Cazaunau, M., Pangui, E., David, M., and Doussin, J.-F.: Photolysis and oxidation by OH radicals of two carbonyl nitrates: 4-nitrooxy-2-butanone and 5-nitrooxy-2-pentanone, Atmos. Chem. Phys., 20, 487-498, https://doi.org/10.5194/acp-20-487-2020, 2020.

Wang, Y., Takeuchi, M., Wang, S., Nizkorodov, S. A., France, S., Eris, G., and Ng, N. L.: Photolysis of Gas-Phase Atmospherically Relevant Monoterpene-Derived Organic Nitrates, J Phys Chem A, 127, 987-999, https://doi.org/10.1021/acs.jpca.2c04307, 2023.

**5. The EVAPORATION and SIMPOL methods are mentioned, but their differences (and potential biases) are not discussed.**

**Response:**

Thank you for your suggestion. The paragraph describing vapor pressure estimation methods is revised and the potential uncertainties of these methods are discussed in the revision:

"*The vapor pressures of the above-mentioned ON species were estimated using two group contribution methods: EVAPORATION (Compernolle et al., 2011) and SIMPOL (Pankow and Asher, 2008). They are both widely used structure activity relationship (SAR)-based group contribution models to predict molecular vapor pressures. The key difference is that EVAPORATION considers proximity-based functional group interactions, so it considers differences in the locations of functional groups, while predictions from SIMPOL do not vary based on functional group locations. As a result, isomeric compounds with the same functional groups but different structures may have different predicted vapor pressures using EVAPORATION but the same using SIMPOL. Therefore, the EVAPORATION model is preferred when chemicals structures are known while the SIMPOL model could be biased. In a recent study, we showed that the EVAPORATION-based kinetic model*

*predicts isoprene SOA more accurately than the SIMPOL-based model, which underpredicts by ~ 20% (Shen et al., 2024).*

*In this work, because the chemical structures of the major ON species are known based on our recent work (Mayorga et al., 2022), we adopted the EVAPORATION method in all our simulations. As the EVAPORATION model input, the structures of the ON species from Mayorga et al. (2022) were converted to SMILES strings. To illustrate the difference between the two models, the EVAPORATION-predicted vapor pressures were compared with SIMPOL predictions (Table S3). The two methods predict vapor pressures within one order of magnitude in most cases, which is typically considered acceptable uncertainties for group contribution vapor pressure estimations.*"
in Line 128 to Line 146.

Reference

Compernolle, S., Ceulemans, K., and Müller, J. F.: EVAPORATION: a new vapour pressure estimation methodfor organic molecules including non-additivity and intramolecular interactions, Atmos. Chem. Phys., 11, 9431-9450, https://doi.org/10.5194/acp-11-9431-2011, 2011.

Pankow, J. F. and Asher, W. E.: SIMPOL.1: a simple group contribution method for predicting vapor pressures and enthalpies of vaporization of multifunctional organic compounds, Atmos. Chem. Phys., 8, 2773-2796, https://doi.org/10.5194/acp-8-2773-2008, 2008.

Shen, C., Yang, X., Thornton, J., Shilling, J., Bi, C., Isaacman-VanWertz, G., and Zhang, H.: Observation-constrained kinetic modeling of isoprene SOA formation in the atmosphere, Atmos. Chem. Phys., 24, 6153-6175, https://doi.org/10.5194/acp-24-6153-2024, 2024.

Mayorga, R., Xia, Y., Zhao, Z., Long, B., and Zhang, H.: Peroxy Radical Autoxidation and Sequential Oxidation in Organic Nitrate Formation during Limonene Nighttime Oxidation, Environ. Sci. Technol., 15337-15346, https://doi.org/10.1021/acs.est.2c04030, 2022.

6. **Central Africa shows the highest ON burden despite not having the highest limonene emissions (Amazon does). The explanation (oxidant competition with isoprene) is buried in the text.**

**Response:**

Thank you for the comments. We have explained why highest limonene-derived ON burnden in Central Africa rather than Amazon. We have updated in the manuscript:

"*The concentration of oxidants is inherently low in Amazon (Fig. S9b-d) and oxidant scavenging in the presence of isoprene with high concentrations greatly reduce the photochemical formation of limonene-derived ON (Mcfiggans et al., 2019). Thus, oxidant competition with isoprene leads to low burden of limonene-derived ON in Amazon despite the highest burden of limonene there.*" in Line 327 to Line 331.

Reference

McFiggans, G., Mentel, T. F., Wildt, J., Pullinen, I., Kang, S., Kleist, E., Schmitt, S., Springer, M., Tillmann, R., Wu, C., Zhao, D., Hallquist, M., Faxon, C., Le Breton, M., Hallquist, A. M., Simpson, D., Bergstrom, R., Jenkin, M. E., Ehn, M., Thornton, J. A., Alfarra, M. R., Bannan, T. J., Percival, C. J., Priestley, M., Topping, D., and Kiendler-Scharr, A.: Secondary organic aerosol reduced by mixture of atmospheric vapours, Nature, 565, 587-593, https://doi.org/10.1038/s41586-018-0871-y, 2019.

**Minor comments:**

1.  **"Limonene has unique structure..." → "Limonene has a unique structure..."**

**Response:**

Thanks for your suggestion. We updated the statement "*Limonene has a unique structure with an endocyclic double bond and an exocyclic double bond*" from Line 62 to Line 63.

2.  **"The chemical mechanism of ON formation could have an influence..." → "The chemical mechanism of ON formation may influence..."**

**Response:**

Thank you for your suggestion. We updated the statement "*The chemical mechanism of ON formation may influence the formation and spatial distribution of ON.*" from Line 401 to Line 402.

3.  **"This difference contributes to the disparity between the global model results and the idealized experimental results from the box model..." → Revise for conciseness: "This discrepancy highlights differences between global-scale dynamics and idealized box model conditions."**

**Response:**

Thank you for your suggestion. We updated the statement "*This discrepancy highlights differences between global-scale dynamics and idealized box model conditions*" from Line 376 to Line 377.

**Special thanks to you for the very constructive comments!**

**Response to the comments from reviewer #2**

**Summary:**

This manuscript addresses a critical gap in atmospheric chemistry modeling by developing and implementing an explicit chemical mechanism for limonene-derived organonitrates (ON) in both box and global models. The incorporation of 90 gas-phase reactions and 39 intermediates represents a substantial advance over simplified schemes, and the sensitivity experiments vividly illustrate nonlinear interactions among OH, $O_3$, and $NO_3$ oxidation pathways. The explicit chemical mechanisms developed here significantly advance the field and offer a robust framework for future studies on secondary organic aerosols. The work is timely, given the increasing recognition of ON's role in secondary organic aerosol formation. I support publication after minor revisions to improve clarity in following comments.

Response to **Overall issues**:

Thanks for your appreciation to our study and constructive comments. We have carefully revised this manuscript based on the reviewer's comments. We hope that the revisions and improvements would satisfactorily address the reviewer's concerns.

**Major comments:**

1. **The introduction effectively contextualizes the importance of ON in SOA and the gaps in current understanding. However, the transition from general SOA/ON to limonene-specific mechanisms could be smoother. Consider briefly mentioning the structural uniqueness of limonene earlier (e.g., around Line 59) to better justify its selection as the focus of this study.**

**Response:**

We appreciate your recommendations about precursor selection in the introduction section. The structural uniqueness of limonene has been mentioned from Line 62 to Line 63. To denote this, we have also emphasized in the abstract section that limonene has a unique structure.

We have updated in the manuscript:

"*However, ON formation from limonene, a major monoterpene with unique structure, and its sensitivity to oxidation pathways remain insufficiently explored due to the absence of models with explicit chemical mechanisms.*" in Line 11 to Line 13.

2. **Lines 73-80: The discussion of model limitations is useful, but it would be helpful to explicitly state how this study addresses these limitations (e.g., by incorporating explicit mechanisms). This could be clarified further.**

**Response:**

Thanks for the comments. An explicit statement addressing the model's limitations is necessary and has been clearly outlined in the introduction section.

We have updated in the manuscript:

"*Thus, incorporating explicit mechanisms is helpful to understand limonene-derived ON formation process and the influence of interaction between multiple*

*oxidation pathways on ON formation.*" in Line 79 to Line 81.

**3. Lines 118: The vapor pressure estimation methods are well-explained, but a brief discussion on the potential uncertainties or limitations of these methods (e.g., sensitivity to molecular structure) would strengthen this section**

**Response:**

Thank you for your suggestion. The paragraph describing vapor pressure estimation methods is revised and the differences and potential biases are discussed in the revision:

"*The vapor pressures of the above-mentioned ON species were estimated using two group contribution methods: EVAPORATION (Compernolle et al., 2011) and SIMPOL (Pankow and Asher, 2008). They are both widely used structure activity relationship (SAR)-based group contribution models to predict molecular vapor pressures. The key difference is that EVAPORATION considers proximity-based functional group interactions, so it considers differences in the locations of functional groups, while predictions from SIMPOL do not vary based on functional group locations. As a result, isomeric compounds with the same functional groups but different structures may have different predicted vapor pressures using EVAPORATION but the same using SIMPOL. Therefore, the EVAPORATION model is preferred when chemicals structures are known while the SIMPOL model could be biased. In a recent study, we showed that the EVAPORATION-based kinetic model predicts isoprene SOA more accurately than the SIMPOL-based model, which underpredicts by ~ 20% (Shen et al., 2024).*

*In this work, because the chemical structures of the major ON species are known based on our recent work (Mayorga et al., 2022), we adopted the EVAPORATION method in all our simulations. As the EVAPORATION model input, the structures of the ON species from Mayorga et al. (2022) were converted to SMILES strings. To*

*illustrate the difference between the two models, the EVAPORATION-predicted vapor pressures were compared with SIMPOL predictions (Table S3). The two methods predict vapor pressures within one order of magnitude in most cases, which is typically considered acceptable uncertainties for group contribution vapor pressure estimations.*"
in Line 128 to Line 146.

Reference

Compernolle, S., Ceulemans, K., and Müller, J. F.: EVAPORATION: a new vapour pressure estimation methodfor organic molecules including non-additivity and intramolecular interactions, Atmos. Chem. Phys., 11, 9431-9450, https://doi.org/10.5194/acp-11-9431-2011, 2011.

Pankow, J. F. and Asher, W. E.: SIMPOL.1: a simple group contribution method for predicting vapor pressures and enthalpies of vaporization of multifunctional organic compounds, Atmos. Chem. Phys., 8, 2773-2796, https://doi.org/10.5194/acp-8-2773-2008, 2008.

Shen, C., Yang, X., Thornton, J., Shilling, J., Bi, C., Isaacman-VanWertz, G., and Zhang, H.: Observation-constrained kinetic modeling of isoprene SOA formation in the atmosphere, Atmos. Chem. Phys., 24, 6153-6175, https://doi.org/10.5194/acp-24-6153-2024, 2024.

Mayorga, R., Xia, Y., Zhao, Z., Long, B., and Zhang, H.: Peroxy Radical Autoxidation and Sequential Oxidation in Organic Nitrate Formation during Limonene Nighttime Oxidation, Environ. Sci. Technol., 15337-15346, https://doi.org/10.1021/acs.est.2c04030, 2022.

4. **Lines 147-149: The global model setup is clearly described, but it would be helpful to briefly justify the choice of CESM/IMPACT over other models, especially given the focus on explicit mechanisms.**

**Response:**

Thanks for your suggestion. The revised manuscript has included the specific advantages of CESM/IMPACT model on incorporating explicit chemical mechanism.

We added "*The CESM/IMPACT model has included a fully explicit gas-phase photochemical mechanism to predict the formation of semi-volatile organic compounds (SVOCs) which then partition to an aerosol phase (Lin et al., 2014), facilitating the*

*incorporation of explicit limonene-derived ON mechanism to simulate their global burden.*" in Line 173 to Line 176.

Reference:

Lin, G., Sillman, S., Penner, J. E., and Ito, A.: Global modeling of SOA: the use of different mechanisms for aqueous-phase formation, Atmos. Chem. Phys., 14, 5451-5475, https://doi.org/10.5194/acp-14-5451-2014, 2014.

5. **The decrease in ON production at high NO$_3$ concentrations (Line 185) is attributed to the dominance of the LIMAL + NO$_3$ pathway (yield: 9.2%). The abrupt transition in Figure 2c (from increase to decrease) warrants a brief discussion of the timescales involved. Is this a kinetic effect (e.g., NO$_3$ outcompeting other pathways) or a thermodynamic limitation?**

**Response:**

Thank you for your comments. We have explicitly showed the timescales under discussion. We update the statement "*When limonene-derived ON concentrations reached steady state within 30 minutes, compared to the case with initial NO$_3$ concentration of $1.0 \times 10^{12}$ molecules cm$^{-3}$, reaction of LIMAL and NO$_3$ become the dominant pathway in the case with initial NO$_3$ concentration of $1.0 \times 10^{17}$ molecules·cm$^{-3}$.*" in Line 214 to Line 217.

The abrupt transition in Figure 2c is a kinetic effect and we have noted this limitation in the manuscript (Line 217 to Line 218). "*The lower yield of the NO$_3$ oxidation pathway (9.2%) of LIMAL relative to OH oxidation pathway (28.8%) results in decreased limonene-derived ON (green box in Fig. S1).*"

6. **Figure 2: The trends in ON production under different oxidant concentrations are clearly presented. However, the discussion of the NO$_3$-initiated pathway**

**(Lines 183-191) could benefit from a more explicit comparison to the OH- and O₃-initiated pathways to highlight the mechanistic differences.**

**Response:**

Thank you for the comments, and we have presented the discussion comparing the NO₃-initiated oxidation pathway with those initiated by OH and O₃ in the manuscript. "*Same to the cases of the OH- and O₃-initiated oxidation pathways, limonene-derived ON increases when initial NO₃ concentrations below $1.0 \times 10^{12}$ molecules·cm$^{-3}$ could be caused by incompletely consumed limonene (Fig. 2c). The increased consumption of limonene with increase in concentrations of NO₃ lead to the increased production of ON. However, different from the cases of OH- and O₃-initiated oxidation pathways, as initial NO₃ concentrations continued to increase, limonene-derived ON production decrease.*" in Line 209 to Line 214.

7. **Figure 2: The y-axis label should specify whether ON concentrations are gas-phase, particle- phase, or total?**

**Response:**

Thank you for your suggestion. We have added the abbreviations of particulate limonene-derived ON to the y-axis in captions of Figure 2-4 to help readers understand the figures.

[Figure]

**Figure 2.** Variations of limonene-derived ON in individual oxidation pathway under different oxidant concentrations. The triangles represent concentration of limonene-derived ON in each experiment. The lines represent the trend of limonene-derived ON. The three datapoint colors represent three initial oxidation pathways (red for OH-initiated oxidation, yellow for $O_3$-initiated oxidation, green for $NO_3$-initiated oxidation).

[Figure]

**Figure 3.** Simulated limonene-derived ON in two initial oxidation pathways under different oxidant conditions, including variation of production of limonene-derived ON with adding (a) initial $O_3$ concentration and (b) initial $NO_3$ concentration in the three OH levels; variation of limonene-derived ON production with adding (c) initial OH concentration and (d) initial $NO_3$ concentration in the three $O_3$ levels; variation of limonene-derived ON production with adding (e) initial OH concentration and (f) initial $O_3$ concentration in the three $NO_3$ levels.

[Figure]

Figure 4. The influence of adding OH-, $O_3$- and $NO_3$-initiated oxidation pathways on the production of limonene-derived ON under different oxidant conditions, including variation of limonene-derived ON production with adding initial OH concentration in the three $O_3$ levels under (a) low, (b) moderate and (c) high $NO_3$ levels; variation of limonene-derived ON production with adding initial $O_3$ concentration in the three OH levels under (d) low, (e) moderate and (f) high $NO_3$ levels; variation of limonene-derived ON production with adding initial $NO_3$ concentration in the three OH levels under (d) low, (e) moderate and (f) high $O_3$ levels. In each panel, the types marked on the columns show the cases when limonene is not completely consumed (type 1) and almost completely consumed (large (type 2) and small (type 3) changes in limonene-derived ON production).

**8. The explanation for low ON burdens in the Amazon (despite high limonene) due to oxidant competition with isoprene (Lines 298-305) is plausible but speculative without quantification. Consider adding a sentence referencing modeled oxidant budgets or prior studies showing isoprene's oxidant sink role.**

**Response:**

Thanks for the suggestion. We have added a sentence referencing prior studies showing oxidant scavenging in the presence of isoprene greatly reduce the photochemical formation of limonene-derived ON.

We have updated in the manuscript:

"*The concentration of oxidants is inherently low in Amazon (Fig. S9b-d) and oxidant scavenging in the presence of isoprene with higher concentrations greatly reduce the photochemical formation of limonene-derived ON (Mcfiggans et al., 2019). Thus, oxidant competition with isoprene leads to low burden of limonene-derived ON in Amazon despite the highest burden of limonene there.*" In Line 327 to Line 331.

Reference

McFiggans, G., Mentel, T. F., Wildt, J., Pullinen, I., Kang, S., Kleist, E., Schmitt, S., Springer, M., Tillmann, R., Wu, C., Zhao, D., Hallquist, M., Faxon, C., Le Breton, M., Hallquist, A. M., Simpson, D., Bergstrom, R., Jenkin, M. E., Ehn, M., Thornton, J. A., Alfarra, M. R., Bannan, T. J., Percival, C. J., Priestley, M., Topping, D., and Kiendler-Scharr, A.: Secondary organic aerosol reduced by mixture of atmospheric vapours, Nature, 565, 587-593, https://doi.org/10.1038/s41586-018-0871-y, 2019.

9. **The 44.7% increase in ON burden from adding OH (Line 330) contrasts sharply with the box model's lower OH-initiated yield (2.1%, Line 195). This discrepancy should be explicitly addressed: Is it driven by regional OH abundance (e.g., tropical OH hotspots) or nonlinear interactions in the global model?**

**Response:**

Thanks for the suggestion. This discrepancy may be caused by nonlinear interactions under complex oxidation conditions in global simulation. We have updated in the manuscript: "*Compared to the simplified condition in simulation using chemical*

*box model, global simulation considers diurnal variations of oxidation. When the O₃-*
*initiated oxidation pathway produces the same amount of limonene-derived ON as the*
*OH-initiated pathway, it consumes more NO₃. As a result, increasing the O₃ oxidation*
*pathway reduces the availability of NO₃ for the nitration of intermediate oxidation*
*products in the nighttime, thereby lowering the total limonene-derived ON yield across*
*all three pathways.*" in Line 365 to Line 370.

**10. Lines 341-345: The nonlinear responses to multiple pathways are well-explained, but a brief mention of how these findings align with or diverge from prior laboratory or modeling studies would provide broader context.**

**Response:**

    Thanks for the suggestion. We have added the similar result from previous research. "*Prior laboratory study has also demonstrated that investigating the response of ON reveals complex and nonlinear behaviour with implications that could inform expectations of changes to ON concentrations as efforts are made to reduce oxidant concentrations (Mayhew et al., 2023).*" in Line 379 to Line 382.

Reference
Mayhew, A. W., Edwards, P. M., and Hamilton, J. F.: Daytime isoprene nitrates under changing NOx and O3, Atmos. Chem. Phys., 23, 8473-8485, https://doi.org/10.5194/acp-23-8473-2023, 2023.

**11. Lines 364-365: A specific example of a missing mechanism or future experimental validation could make this more concrete. Are there missing pathways (e.g., heterogeneous NO₃ reactions) that could alter conclusions?**

**Response:**

    Thanks for the comment. We have showed the missing mechanisms in our work,

and the missing pathways could not alter conclusions in the manuscript. "*We only include main oxidation process published to date in the model, while some pathways (i.e. Heterogeneous NO$_3$ reactions) of ON is missing in this work. Gas-phase oxidation in our mechanism is considered as the dominant formation pathway of ON (Fan et al., 2022; Perring et al., 2013). Future inclusion of newly identified and quantified ON chemistry may lead to unpredictable nonlinear impacts on simulation outcomes.*" in Line 402 to Line 406.

Reference

Fan, W., Chen, T., Zhu, Z., Zhang, H., Qiu, Y., and Yin, D.: A review of secondary organic aerosols formation focusing on organosulfates and organic nitrates, J. Hazard. Mater., 430, 128406, https://doi.org/10.1016/j.jhazmat.2022.128406, 2022.

Perring, A. E., Pusede, S. E., and Cohen, R. C.: An Observational Perspective on the Atmospheric Impacts of Alkyl and Multifunctional Nitrates on Ozone and Secondary Organic Aerosol, Chem. Rev., 113, 5848-5870, https://doi.org/10.1021/cr300520x, 2013.

**12. The implications for policy or air quality management could be expanded slightly, given the anthropogenic-biogenic interaction focus**

**Response:**

Thanks for the suggestion. The implications for policy and air quality management implications have been expanded in the revision:

"*Quantitative understanding of these complex interactions in contributing to SOA formation can definitely facilitate better understanding the contributions of interactions and antagonistic actions between anthropogenic and natural emissions to atmospheric aerosols. These works provide valuable insights for making more effective secondary aerosol pollution control strategies to improve air quality.*" in Line 411 to Line 415.

**Minor comments:**

1. **Line 47: "evaded" should likely be "avoided".**

**Response:**

Thank you for the comments, and we apologize for the ambiguous expression. The statement has been corrected in Line 52 to Line 53 "*Compared with single oxidant, the introduction of multiple oxidants leads to the possible complex reaction mechanisms for VOCs.*".

2. **Line 104: "limonaldehyde" → "limononaldehyde" (consistency with MCM).**

**Response:**

Thanks for your suggestion. We updated the statement "*The rate constants were set to be the same as those used in MCM for limononaldehyde.*" from Line 107 to Line 108.

3. **Line 132: "$1.0\times10^{11}$ molecules·cm$^{-3}$" seems high for limonene; consider clarifying if this is a typo or based on specific experimental conditions.**

**Response:**

Thanks for your suggestion and we have updated "*Limonene at a concentration of $1.0\times10^{11}$ molecules·cm$^{-3}$ was used as the precursor for ON, which falls within the range of values reported in laboratory and observation studies (Guo et al., 2022; Luo et al., 2023; Ham et al., 2016).*" from Line 154 to Line 156 to show the references of the limonene concentration setting.

Reference:

Guo, Y., Shen, H., Pullinen, I., Luo, H., Kang, S., Vereecken, L., Fuchs, H., Hallquist, M., Acir, I.-H., Tillmann, R., Rohrer, F., Wildt, J., Kiendler-Scharr, A., Wahner, A., Zhao, D., and Mentel, T. F.: Identification of Highly Oxygenated Organic Molecules and Their Role in Aerosol Formation in the Reaction of Limonene with Nitrate Radical, Atmos. Chem. Phys., 22, 11323-11346, https://doi.org/10.5194/acp-22-11323-2022, 2022.

Luo, H., Vereecken, L., Shen, H., Kang, S., Pullinen, I., Hallquist, M., Fuchs, H., Wahner, A., Kiendler-Scharr, A., Mentel, T. F., and Zhao, D.: Formation of Highly Oxygenated Organic Molecules from the Oxidation of Limonene by OH Radical: Significant Contribution of H-abstraction Pathway, Atmos. Chem. Phys., 23, 7297-7319, https://doi.org/10.5194/acp-23-7297-2023, 2023.

Ham, J. E., Harrison, J. C., Jackson, S. R., and Wells, J. R.: Limonene ozonolysis in the presence of nitric oxide: Gas-phase reaction products and yields, Atmos Environ (1994), 132, 300-308, https://doi.org/10.1016/j.atmosenv.2016.03.003, 2016.

**4. Line 224: "phenomena" should be "phenomenon".**

**Response:**

Thanks for your suggestion. We updated the statement "*A similar phenomenon observed in laboratory study shows that NO$_x$ influences γ-terpinene ozonolysis by enhancing NO$_3$ production at high NO$_x$ levels.*" from Line 254 to Line 256.

**Other minor changes according to reviewer's comments:**

1. We have corrected the mean value of Figure S10c in the Supplement.

[Figure]

**Figure S10.** The difference in column concentration of limonene-derived ON between Case3 and Case4 (a), the difference in column concentration of limonene-derived ON between Case2 and Case4 (b), and the difference in column concentration of limonene-derived ON between Case1 and Case5 (c), the difference in column concentration of limonene-derived ON between Case3 and Case5 (d), the difference in column concentration of limonene-derived ON between Case1 and Case6 (e), the difference in column concentration of limonene-derived ON between Case2 and Case6 (f).

**Once again, thank you very much for all the comments and suggestions.**